# Peri-Interventional Triple Therapy with Dabigatran Modifies Vasomotion after Bare-Metal Stent Implantation in a Pig Coronary Artery Model

**DOI:** 10.3390/jpm13020280

**Published:** 2023-01-31

**Authors:** Rayyan Hemetsberger, Serdar Farhan, Dominika Lukovic, Katrin Zlabinger, Judit Hajagos-Toth, Judit Bota, Hector M. Garcia-Garcia, Cihan Ay, Eslam Samaha, Robert Gaspar, Rita Garamvölgyi, Kurt Huber, Andreas Spannbauer, Mariann Gyöngyösi

**Affiliations:** 1Department of Cardiology, 2nd Department of Internal Medicine, Medical University of Vienna, 1090 Vienna, Austria; 2Zena and Michael A. Wiener Cardiovascular Institute, Icahn School of Medicine at Mount Sinai, New York, NY 10029, USA; 3Department of Pharmacology and Pharmacotherapy, Albert Szent-Györgyi Medical School, University of Szeged, 6720 Szeged, Hungary; 4Medstar Hospital Center, Department of Cardiology, Washington, DC 20010, USA; 5Clinical Division of Haematology and Haemostaseology, Department of Medicine I, Medical University of Vienna, 1090 Vienna, Austria; 6Institute of Diagnostics and Radiation Oncology, University of Kaposvar, 7400 Kaposvar, Hungary; 73rd Medical Department with Cardiology, Wilhelminen Hospital, 1160 Vienna, Austria

**Keywords:** vasomotor function, neointimal formation, dabigatran, preclinical, bare metal stent

## Abstract

(1) Background: Coronary artery stenting leads to local inflammation, disturbs vasomotion, and slows endothelialization, increasing vascular thrombus risk. We used a pig stenting coronary artery model to assess how peri-interventional triple therapy with dabigatran ameliorates these effects. (2) Methods: In a total of 28 pigs bare-metal stents were implanted. Four days before the percutaneous coronary intervention (PCI), we started 16 of the animals on dabigatran, maintained through 4 days after the procedure. As controls, the remaining 12 pigs received no therapy. In both groups, dual antiplatelet therapy (DAPT) (clopidogrel, 75 mg plus aspirin, 100 mg) was administered until animals were euthanized. Just after the PCI and on day 3 after the procedure, we performed optical coherence tomography (OCT) in eight animals in the dabigatran group and four controls and euthanized them. We followed the eight remaining animals in each group with OCT and angiography for one month before euthanizing them and performed in vitro myometry and histology on harvested coronary arteries from all animals. (3) Results: The dabigatran group showed a significantly increased vasoconstriction at 3 days after PCI (10.97 ± 3.85 mN vs. 7.32 ± 5.41 mN, *p* = 0.03), but we found no differences between endothelium-dependent and -independent vasodilatation. We also found no group differences in OCT, quantitative angiography, or histomorphometry findings. (4) Conclusions: Starting a short course of dabigatran just before PCI and continuing for a 3-day window along with usual post-PCI DAPT is associated with enhanced vasoconstriction after bare-metal stent implantation without reducing neointimal formation at one month.

## 1. Introduction

Treatment of stenotic coronary artery disease with a metallic coronary stent is considered the gold standard for symptomatic patients [1]. This percutaneous coronary intervention (PCI) induces vascular injury, however, leading to local inflammation, altered vasomotion, and enhanced neointimal hyperplasia [2]. To address the limitation of in-stent re-stenosis, drug-eluting stents (DES) were introduced to inhibit in-stent cellular proliferation [2]. Furthermore, treatment with dual antiplatelet therapy (DAPT) is required to prevent thrombotic events during the process of endothelialization over the stent struts. Adding an anticoagulant to DAPT in patients at risk for atherothrombotic events is linked to better ischemic outcomes compared with using either as monotherapy, but bleeding risk is increased [3,4].

We hypothesized that a dabigatran-related reduction in thrombin levels would lead to dampened inflammatory activation after local vascular injury. Using a pig coronary artery model, we previously showed that initiating triple anticoagulant therapy with aspirin, clopidogrel, and dabigatran just before DES implantation and continuing the therapy for a short time afterward resulted in improved endothelium-dependent vasodilatation, faster strut endothelialization, and reduced expression of MCP-1 [5]. However, the short-term periprocedural triple therapy showed no effect on neointimal proliferation in the longer term, at one month. 

Following on that study, we sought to assess whether the results were attributable to dabigatran alone or to the effects of the antiproliferative drug on the DES, as well. For this purpose, we analyzed the effect of triple therapy (dabigatran plus DAPT) initiated just before and continued for a short time after coronary stenting with bare metal stents (BMS) in a pig coronary artery model.

## 2. Materials and Methods

### 2.1. Drug Dosing

On the day before the intervention, juvenile domestic pigs (*n* = 28) with an average weight of 32.6 ± 2.3 kg were orally administered a loading dose of DAPT (clopidogrel, 300 mg plus aspirin, 100 mg). Animals then received daily DAPT (clopidogrel 75 mg plus 100 mg aspirin) until they were euthanized (see below). Of the 28 pigs, 16 were randomized to received triple therapy (the addition of dabigatran to DAPT), and the remaining 12 served as controls, receiving only DAPT. Dabigatran (20 mg/kg dabigatran given orally twice daily, or about 600 mg two times a day, similar to a human dose of 2 × 150 mg) was begun 4 days before stenting to ensure a steady-state serum level, as pigs can have delayed or decreased absorption of this drug [6]. The dabigatran group also continued with triple therapy to day 4 after the intervention because previous work has shown detection of MCP-1 through this period following blood vessel injury [7] (Figure 1).

For anesthesia, we began with 0.04 mg/kg atropine, 12 mg/kg ketamine-hydrochloride, and 1 mg/kg xylazine, deepening the effect with facemask administration of isoflurane and O_2_. After animals were intubated, ongoing anesthesia consisted of a mixture of isoflurane (2–3.5 vol%), O_2_ (1.6–2.5 vol%), and N_2_O (0.5 vol%). All animals were continuously monitored for blood pressure and O_2_ saturation and with electrocardiography. After surgical preparation of the right femoral artery to allow insertion of a 7F introducer sheath (Radifocus Introducer II, Terumo Medical Corporation, Somerset, NJ, USA), we performed arteriotomy under sterile conditions. Unfractionated heparin was administered during the procedure based on activated clotting time (range, 200–300 s). 

The Ethical Committee on Animal Experiments of the University of Kaposvar, Hungary, approved the study based on accepted animal care and use principles (NIH publication No. 86-23, revised 1985). The study was conducted in the Institute of Diagnostics and Oncoradiology, University of Kaposvar, Hungary.

### 2.2. PCI

After a 7F guiding catheter (Medtronic, Minneapolis, MN, USA) was introduced into the left coronary artery ostium, coronary angiography with a nonionic contrast agent was performed. To place the stent, a guiding wire (Cordis, Miami Lake, FL, USA) was introduced into the left anterior descending and left circumflex coronary arteries. The stainless steel tubular stent with a low strut thickness [8] (80 µm; Tsunami, Terumo Corp, Tokyo, Japan) (diameter, 3.2 ± 0.3 mm; length 20.6 ± 5.5 mm) was implanted into one or the other, alternately, following previously published guidelines [9]. After stents were placed, intracoronary imaging was performed using intravascular OCT (C7-XR imaging system; St. Jude Medical, LightLab Imaging, Inc, Westford, MA, USA), with an image catheter (Dragonfly™, St. Jude Medical, LightLab Imaging, Inc.) positioned distal to the stent. Contrast (4 mL/s) was flushed continuously through the guiding catheter for blood clearance with motorized pullback. To capture and digitally store all images for later analysis offline, we used the ILUMIEN System (St. Jude Medical, St. Paul, MN, USA). After the procedure, the arteriotomy was ligated, the skin closed in two layers, and animals allowed to recover.

### 2.3. Follow-Up

Day 3 was selected for the first follow-up for two reasons. First, the first 4 days after PCI represent the window of risk for the highest thrombus burden [10]. Second, upregulation of MCP-1 expression is seen only during this initial window after vascular injury [7,11,12]. A second (and final) follow-up took place at one month, which is similar to follow-up at 6 months in humans [9,13,14]. Throughout all follow-up periods, animals received DAPT, with dabigatran added for the triple-therapy group for the first 4 days after stent placement. We performed coronary angiography and OCT at follow-up. For the first follow-up at 3 days, we euthanized eight pigs from the triple-therapy group and four from among controls. The remaining eight animals in each group were euthanized at the second follow-up (one month). All euthanasia was conducted using saturated potassium-chloride (10 mL), after which hearts with stented arteries were explanted for histological and myometric analyses.

In preparation for these studies, the coronary vessels were dissected carefully and cut into segments (containing the stent and proximal and distal to it). To flush segments, we used physiological saline.

For in vitro studies of vasomotor response, the segments proximal and distal to the stent were cut into 4-mm rings and analyzed immediately. As an unstented comparator, we used a 4-mm ring taken from the untreated right coronary artery (RCA).

To prepare day 3 samples for histomorphometric and histopathological analyses, we cut the stented segments into halves and fixed one in 4% formalin, following by embedding in Technovit 9100. We placed the other half in RNAlater for 24 h at 4 °C and then stored it at −20 °C for quantitative real-time PCR (qPCR). Samples from the second, one-month follow-up also were embedded in Technovit 9100 for histology, then cut into slices 4–6 µm thick, followed by staining (hematoxylin–eosin, MOVAT pentachrome).

### 2.4. Analyses

For all analyses, treatment group was masked.

#### 2.4.1. Quantitative Coronary Angiography

For quantitative coronary angiography after stent placement and at follow-up, we used a computer-assisted quantitative arteriographic edge-detection algorithm (ACOM.PC, Siemens, Erlangen, Germany).

#### 2.4.2. OCT

To perform OCT measurements, we used LightLab Imaging software, as previously described [15]. To quantify the peri-strut tissue burden, i.e., structures on the struts most likely to be a thrombus and/or fibrin, we used a semi-quantitative score [5].

#### 2.4.3. Histomorphometry and Histopathology

For the specifics of the histopathology and histomorphometry analyses, see the Appendix A.

#### 2.4.4. qPCR

We used qPCR to quantify MCP-1 and PAR-1 levels in the stented segments, with stent-free RCA as control. Tissue first was placed in RNAlater (Qiagen, Hilden, Germany) and then mRNA reverse transcribed to cDNA with QIAGEN miScript RT kit (Qiagen, Hilden, Germany). To quantify expression, we used qPCR with miScript SYBR® Green PCR Kit (Qiagen, Hilden, Germany) on an Applied Biosystems 7500 Real-Time PCR System (Life Technologies, Carlsbad, CA, USA). We selected MCP-1 and PAR-1 because of their suspected involvement in the development of in-stent restenosis. Target sequence primers were designed with Primer3 (http://primer3.wi.mit.edu/primer3web_help.htm accessed on 26 January 2022; Microsynth, Switzerland). To normalize target gene expression rates, we used the averages of stably expressed housekeeping genes (ACTB, HPRT1, PPIA) as endogenous controls. We calculated both relative gene expression level per the ΔΔCt method and changes relative to expression levels detected in the RCA samples.

#### 2.4.5. Vasomotor Responses

A stent is a rigid metallic cage that has been reported to trigger changes in coronary artery vasomotor function relative to systolic-diastolic movement, carrying the risk of increased shear stress and resulting complications [16]. Thrombin inhibition and the consequent disruption of thrombus formation and the inflammation cascade in the few days following stenting could affect vasomotor function. To assess for such effects, we used the 4-mm arterial rings from the stent-free proximal and distal segments, first removing adhesive fat and connective tissues. Segments were mounted in a 15-mL tissue bath (37 °C) of modified Krebs–Hensel it buffer solution (K3753, Sigma-Aldrich, Vienna, Austria) [17,18]. We maintained the buffer with fresh solution supplied from a reservoir dish via peristaltic pump (Ismatec REGLO Digital, IDEX, Germany) to ensure a steady-state flow of 1.5 mL/min. The reservoir contents were refreshed with a 95% O_2_ and 5% CO_2_ mixture at pH 7.4. 

We suspended the segments between 2 L-shaped metal pins (diameter, 0.4 mm) in a myograph (Living Systems Instrumentation, Catamount R&D Inc, Saint Albans, VT, USA) to measure coronary vessel isometric circular wall tension (Figure 2 and [17]). After being allowed to stabilize for ~1 h, the segments were contracted to the maximum (milliNewtons, mN) with endothelin-1 (30 nM/L). To achieve peak endothelium-dependent vasodilation, we placed substance P (1 nM/L) into the bath. Smooth muscle sensitivity to external nitric oxide (NO) was assessed with exposure in the bath to sodium nitroprusside (4 mM/L). Vasodilation (vascular smooth muscle stiffness after PCI injury), whether endothelium-dependent or not, is given as percent change of steady-state level contraction (mN/s/mN units). The vasomotion experimental setup is visualized in our previous experiment [17] (Figure 2).

#### 2.4.6. Statistics

Results of analyses of continuous parameters are given as means ± standard deviation (SD). Student’s *t*-test or the nonparametric Mann-Whitney U test was used to evaluate between-group differences. All analyses were performed with SPSS, version 23.0 (SPSS, IBM Corporation, NY, USA), with *p* < 0.05 considered to indicate statistical significance.

## 3. Results

### 3.1. Measurements

#### 3.1.1. Quantitative Coronary Angiography

We found no significant difference between groups in quantitative coronary angiography parameters at any time point (baseline, after PCI, day 3, one month; Appendix A).

#### 3.1.2. OCT

The groups also did not differ in OCT parameters (Appendix A). Stent strut burden was statistically lower just after PCI in dabigatran versus control animals, but had declined by day 3 (Table 1, Figure 3).

#### 3.1.3. Histomorphometry and Histopathology

We found no differences between the triple-therapy and DAPT-only groups for any histomorphometric or histopathological parameters (Appendix A), including inflammation, injury score, or fibrin deposition at the day 3 and one-month follow-ups. Both groups showed complete endothelialization by one month (Table 2).

#### 3.1.4. Vasomotor Edge Response

Compared with controls, the dabigatran group showed a greater capacity for endothelin-induced vasoconstriction at day 3. The groups were similar for endothelium-dependent and -independent vasodilation (Table 3).

#### 3.1.5. qPCR Results for MCP-1, PAR-1

In the dabigatran group at day 3, which was during triple-therapy treatment, in-stent tissue MCP-1 expression was significantly upregulated after implantation. However, we found no differences in PAR-1 expression between the treatment groups at day 3 (Table 4).

## 4. Discussion

Here we investigated how dabigatran as part of triple therapy affects the levels of coronary artery inflammation and course of vessel healing after PCI with BMS. Our main findings were an association of peri-interventional dabigatran with increased post-PCI vasoconstriction, along with neointimal proliferation and peri-strut inflammation comparable to controls.

In our previous experiment using the Nobori Biolimus A9–eluting DES, treatment with dabigatran resulted in enhanced endothelium-dependent vasodilatation without relevant effects on vasoconstriction or on endothelium-independent vasodilatation [5]. Given this finding, in the absence of Biolimus A9, i.e., when using a BMS, dabigatran treatment has different effects on vasomotion. Moreover, in both analyses, the change in vasomotion among dabigatran-treated animals was present only after stent implantation. In the control coronary arteries (i.e., RCA), dabigatran did not significantly affect vasomotion. Only after vascular injury in combination with a BMS was vasoconstriction enhanced, whereas in combination with DES, endothelium-dependent vasodilatation was enhanced. Of note, in dabigatran-treated animals, BMS implantation resulted in upregulation of the proinflammatory MCP-1 in the stented tissue as compared with animals not treated with dabigatran. This pattern is different from the previously described downregulation of MCP-1 in DES-implanted tissue of dabigatran-treated animals [5]. However, in both scenarios—after BMS or DES implantation—peri-strut inflammation as evaluated on histopathology did not significantly differ between dabigatran-treated and control animals. The variations in MCP-1 expression could explain the variations in vasomotor reaction with the different stent platforms. 

Paclitaxel-eluting DES has been associated with longer-term impaired vasomotor function in a pig coronary artery model [17,19]. The proneness to endothelium-dependent vasodilatation was reduced in paclitaxel-eluting DES, but the same was not seen with BMS and paclitaxel-eluting balloon–treated animals [16]. An earlier experiment yielded similar results with paclitaxel-eluting DES, but histological markers of inflammation were more pronounced. In agreement with previous reports, we found with BMS that histological markers of inflammation did not differ significantly between dabigatran-treated and control animals. In the same vein, in our previous experiment with a Biolimus A9–eluting DES [5], histological inflammatory markers did not significantly change with use of dabigatran. However, the downregulation of MCP-1 in dabigatran-treated animals – and most probably the favorable vasomotor reaction, as a result—may be explained by the additional use of a novel DES platform.

Although the use of BMS is obsolete in modern PCI [1] and these stents have been completely replaced by newer generation DES, in low-resource regions, they are still implanted because DES is cost-prohibitive. Thus, in addition to the academic interest in the effect of a triple therapy in the absence of a stent-eluted drug, this scenario remains a reality in some countries. As the population ages and the incidence of atrial fibrillation consequently increases, PCI in patients requiring oral anticoagulation will increase as well. Although this preclinical experiment certainly cannot be directly translated into clinical practice, our data may suggest that BMS should not be combined with triple anticoagulatory therapy given an apparent proneness to vasoconstriction, whereas our previous finding supports use of a DES in this clinical setting, given the enhanced endothelium-dependent vasodilatation we observed. 

Limitations. This is an experimental study, so the findings require caution with interpretation and should be considered hypothesis-generating. We used young, healthy pigs for these studies, with arteries that likely did not reflect the atherosclerotic human condition. In addition, juvenile pig arteries endothelialize rapidly after injury. Despite these limitations, the similarities between human and pig arteries are sufficient to warrant use of the latter in modeling PCI outcomes, and they are an established tool for such studies [9].

## 5. Conclusions

We found that peri-interventional triple therapy (dabigatran added to DAPT) in the short term generated enhanced vasoconstriction after bare-metal stent implantation without reducing neointimal formation at one month after PCI.

## Figures and Tables

**Figure 1 jpm-13-00280-f001:**
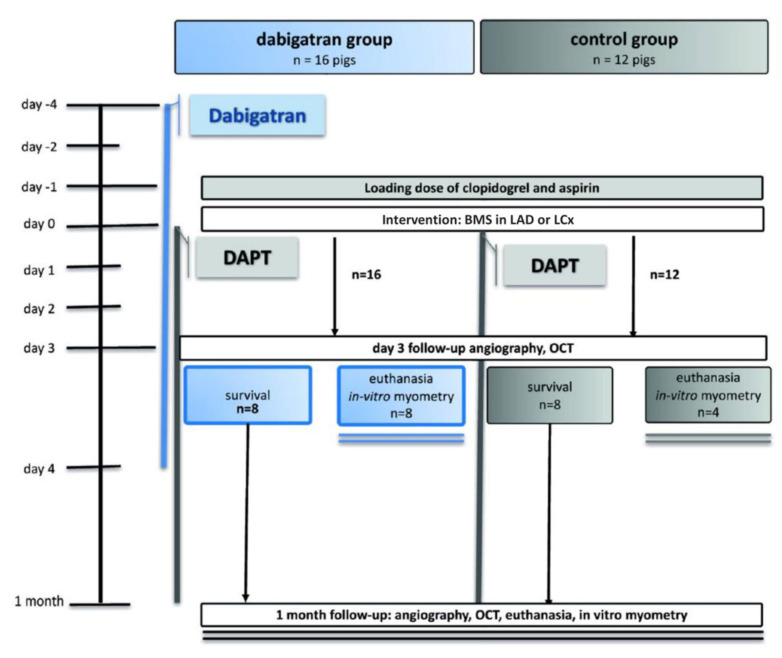
Study design. LAD: left anterior descending coronary artery; LCx, left circumflex artery; BMS: bare metal stent.

**Figure 2 jpm-13-00280-f002:**
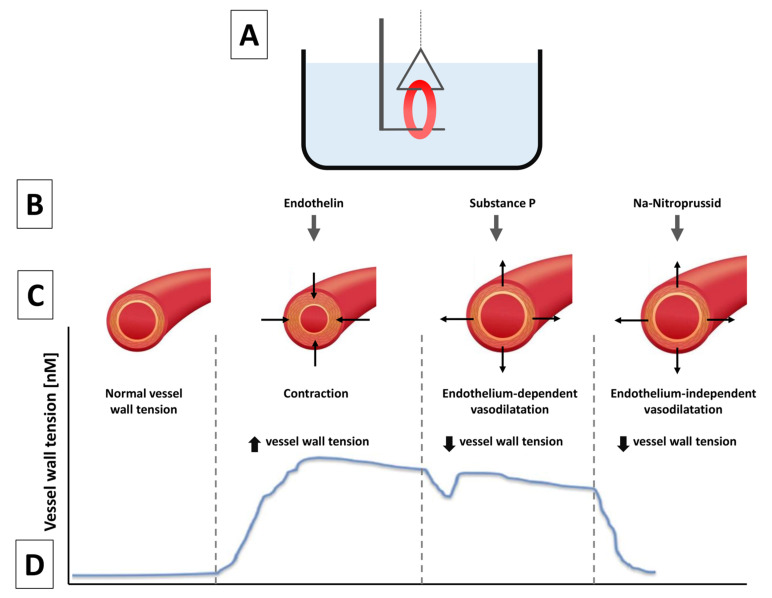
Illustration of In Vitro Measurements of Coronary Vasomotor Reaction (**A**) Schematic presentation of a bath chamber with a myograph and its L-shaped pins. A coronary vessel is mounted on the pins. (**B**) Drugs added into the bath chamber. (**C**) Schematic of the coronary arteries and their response to different compounds. (**D**) Representation of coronary vessel wall tension in response to the different compounds.

**Figure 3 jpm-13-00280-f003:**
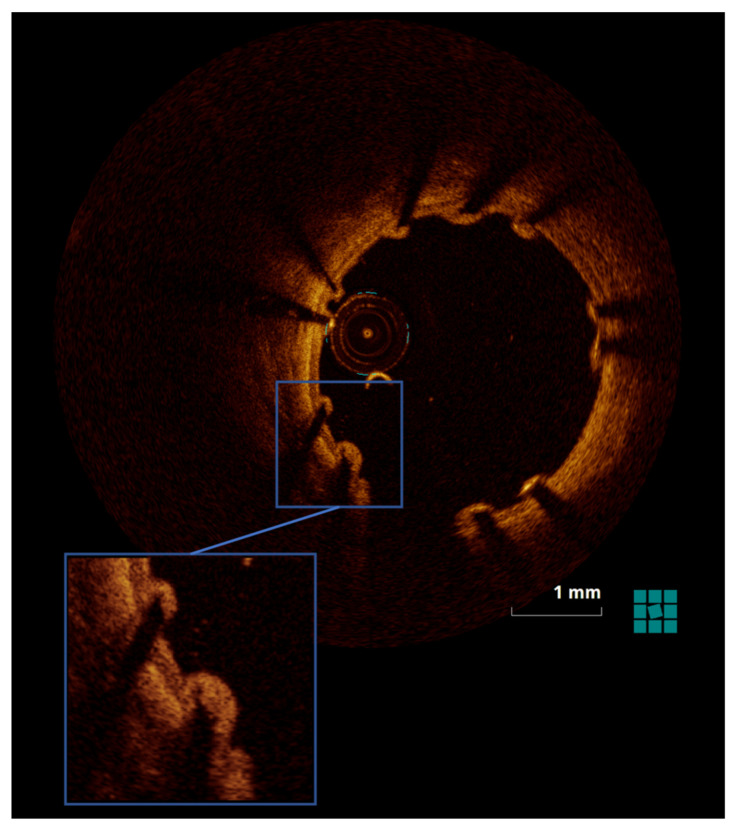
Illustration of tissue burden on stent struts as seen via OCT at day 3 after PCI.

**Table 1 jpm-13-00280-t001:** Tissue burden score per OCT just after PCI and at day 3.

	Immediately after PCI	Day 3 after PCI
**Dabigatran group**	3.17 ± 4.54	5.81 ± 3.27
**Control group**	11.60 ± 10.64	3.63 ± 1.41
** *P* **	0.044	0.120

Mean ± SD.

**Table 2 jpm-13-00280-t002:** Histopathologic results at day 3 and at one month after PCI.

	Inflammation		Fibrin Deposition		Endothelialization (%)	
	Day 3	1 Month	Day 3	1 Month	Day 3	1 Month
**Dabigatran group**	0.55 ± 1.01	1.67 ± 1.86	1.44 ± 0.73	0.50 ± 0.84	38.89 ± 30.90	100 ± 0
**Control group**	1.00 ± 1.15	0.38 ± 0.52	1.75 ± 0.50	0.88 ± 0.64	25.00 ± 20.41	100 ± 0
** *P* **	0.48	0.16	0.47	0.26	0.50	1.0

Mean ± SD.

**Table 3 jpm-13-00280-t003:** Vasomotor edge responses. Vasomotor edge responses after addition of endothelin to measure vasoconstriction, addition of substance P to measure endothelium-dependent vasodilatation, and addition of Na-nitroprusside to induce endothelium-independent vasodilatation in control unstented RCA and in coronary arteries bearing a BMS at day 3 and one month of follow-up.

	Day 3 Non-PCI Coronary	Day 3 BMS	1 Month BMS
**Endothelin-induced constriction [mN]**			
**Dabigatran group**	11.18 ± 3.95	10.97 ± 3.85	9.03 ± 4.94
**Control group**	9.01 ± 2.46	7.32 ± 5.41	8.10 ± 3.14
* **P** *	0.54	0.03	0.45
**Endothelium-dependent vasodilatation (substance P–induced vasodilatation [%])**			
**Dabigatran group**	78 ± 18	63 ± 28	59 ± 30
**Control group**	60 ± 34	72 ± 50	57 ± 29
* **P** *	0.07	0.53	0.82
**Endothelium-independent vasodilatation (Na-nitroprusside–induced vasodilatation [mN/s/mN])**			
**Dabigatran group**	0.15 ± 0.09	0.081 ± 0.047	0.15 ± 0.11
**Control group**	0.13 ± 0.06	0.097 ± 0.033	0.14 ± 0.06
* **P** *	0.42	0.33	0.47

Mean ± SD.

**Table 4 jpm-13-00280-t004:** qPCR analysis of MCP-1 and PAR-1 expression of stented tissue 3 days after bare-metal stenting in dabigatran and control groups.

	MCP-1 (Log Fold Change)	PAR-1 (Log Fold Change)
**Dabigatran group**	0.84 ± 0.23	−0.24 ± 0.19
**Control group**	0.14 ± 0.50	−0.27 ± 0.31
** *P* **	0.045	0.84

Mean ± SD.

## Data Availability

Not applicable.

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
