# Peer review of "Peri-Interventional Triple Therapy with Dabigatran Modifies Vasomotion after Bare-Metal Stent Implantation in a Pig Coronary Artery Model"

_jpm, 2023, doi:10.3390/jpm13020280_

Round 1

Reviewer 1 Report

The authors present a novel experimental study investigating the potential association of triple therapy with impaired vasomotion after bare-metal stenting.

The following issues need to be addressed:

- Methods: Details about the pig cohort are necessary. Were these hypercholesterolemic, diabetic or both ? What was the age of the pigs and how much did they weigh on average?

- Methods: Why did the authors use RCA as control, and not the unstented LAD or Cx accordingly? That could be a better control since it would also be in the left coronary artery

- Methods: Has the dose for dabigatran in pigs been previously standardized?

- Methods - Statistics: Was the normality of continuous variables tested? Is the student's t-test appropriate? For instance, comparing tissue burden score by t-test may not be appropriate

- Discussion: what is the putative mechanism which could explain the association observed? That should be further explored in the discussion section

Author Response

We thank the reviewer for his/her valuable comments and the all over positive evaluation of our work. 

Here we address the comments point-by-point:

Methods: Details about the pig cohort are necessary. Were these hypercholesterolemic, diabetic or both ? What was the age of the pigs and how much did they weigh on average?

The animals were juvenile domestic pigs, not hypercholesterolemic nor diabetic. 

Accordingly we inserted in the manuscript in the methods section - drug dosing the following:

"On the day before the intervention, juvenile domestic pigs (n = 28) with an average weight of 32.6±2.3 kg were orally administered a loading dose of DAPT..." 

Methods: Why did the authors use RCA as control, and not the unstented LAD or Cx accordingly? That could be a better control since it would also be in the left coronary artery

We chose the RCA in order to use an competely 'untouched' vessel.

Methods: Has the dose for dabigatran in pigs been previously standardized?

The dabigatran dose of 20 mg/kg in pigs was previously standardized and can be found in reference no 6 (McKellar SH, Abel S, Camp CL, Suri RM, Ereth MH, Schaff HV. Effectiveness of dabigatran etexilate for thromboprophylaxis of mechanical heart valves. J Thorac Cardiovasc Surg 2011;141:1410-6.)

Methods - Statistics: Was the normality of continuous variables tested? Is the student's t-test appropriate? For instance, comparing tissue burden score by t-test may not be appropriate

We thank the reviewer for this important comment. Accordingly, we controlled the results and tested for normality of continuous variables. Indeed, t-test was not appropriate comparing tissue burden.

We changed the statistic section and corrected the p-values in cases of not normally distributed variables. In the cases where one group was not normally distributed and the other group was normally distributed, both p-values did not change the significance of comparison.

"Student’s t-test or the nonparametric Mann-Whitney U test was used to evaluate between-group differences. "

Discussion: what is the putative mechanism which could explain the association observed? That should be further explored in the discussion section

We hypothese, that the up-regulation of MCP-1 after injury in BMS stented tissue in the dabigatran treated animals may have caused the enhanced vasoconstriction (in absence of cytostatic agents from DES). This was discussed in the discussion section.

Reviewer 2 Report

Dear Authors,

Congratulations to your meticulous work. It's objective is clearly stated and methods and results are formulated flawlessly. You could even provide clinical impact of your findings by suggesting not to use BMS with triple therapy.

Phrasing and English is perfect. Please check tabulating.

Author Response

We thank the reviewer for his/her positive evaluation of our work. We checked tabulating and set the table 4 the groups in bold.